# Family-Size Effect on Intergenerational Income Mobility under China's Family Planning Policy: Testing the Quantity–Quality Trade-Off

**Xinxin Mu [1],\* and Shenghu Chen [2]**

[1] School of Marxism, Liaoning University, Shenyang 110036, China
[2] Institute of Metal Research, Chinese Academy of Sciences, Shenyang 110016, China
\* Correspondence: mx-916@163.com

**Abstract:** The causal link between family size and human capital investment in children is critical for family planning policy. However, empirical studies aiming to test the quantity–quality trade-off are far from sufficient. This paper tried to estimate the family-size effect on intergenerational income mobility using China Family Panel Survey (CFPS) data. The empirical model of intergenerational income mobility with respective to family size was formulated, and the fertility rates allowed by family planning policy were used as an instrument variable for family size. It was found that intergenerational income elasticity tended to decrease with an increase in family size. The impact of family size on intergenerational income elasticity was sensitive to the income rank positions, and nonlinearity in intergenerational transmission of income under unequal family was observed. A quantity–quality trade-off analysis was applied to further test the family-size effect. Pronounced family-size effects were observed in low-income regions with tight budget constraints and in regions with less-developed credit markets, followed by an obvious quantity–quality trade-off. The sex difference in intergenerational transmission of income may be attributed to the existence of the "preference to sons over daughters" phenomenon. The present work provides a theoretical basis for shaping family planning policies toward sustainability.

**Keywords:** family planning policy; family size; intergenerational income mobility; quantity–quality trade-off





## 1. Introduction

Correlation between family size and human capital investment in children has been widely studied by researchers in the fields of economics and demography, and this argument has become one of the important factors for family planning policies in many countries. The quantity–quality model developed by Becker and his co-authors [1,2] suggests that there is a negative relationship between thee human capital investment in child and number of children in the family, which consequently leads to a trade-off between the quantity and quality of children. Extensive studies, especially in Western countries, have been devoted to exploring the relationship between the quantity of children and individual quality through empirical research. This trade-off model has generally been accepted for many years [3–5]. However, recent research findings obtained in different countries have shown large variations and can even be conflicting. Black et al. [6] found that the impact of family size on the quality of children was negligible after considering the birth order of children according to Norwegian national population data. In Israel, evidence of a trade-off between quantity and quality was not found in the non-one-child families by Angrist et al. [7], if the fertility variables were considered. Investigations on the impact of family size on children's educational attainment in Sweden by Åslund and Grönqvist [8] revealed that there was no significant correlation between quantity and quality. Aside from the inherent differences between databases across countries, different situations that arise

in developed and developing countries are responsible for the inconsistent findings. For instance, Norway has established a good public education system and can provide generous support in childhood care, medical treatment and education by the government. In other words, parents bear a small amount of the child's expenses, resulting in weak evidence of a quantity–quality trade-off in developed countries. However, in developing countries such as India, where a comprehensive public education system has not established and cannot provide adequate financial support by government, there is strong possible evidence of this trade-off [9]. In Senegal and Uganda, the existence of this trade-off was found for most of the quality attributes covering education, nutrition, and health care [10]. In Indonesia, statistically meaningful evidence of this trade-off for body mass index at select quantiles in the tails of the distribution was found [11]. Therefore, the relationship between family size and human capital investment in children in developing countries should be given more attention.

Family fertility decisions may be affected by the parents' educational level and preferences, so endogeneity in family size poses a challenge for empirical research. The instrumental variable (IV) method is often used to simultaneously test and control for endogeneity, and twin status is often used as IV for family size. Using multiple births as an exogenous shift in family size, Rosenzweig and Wolpin [12] found that the average education levels were much lower for children from larger families in India. Angrist et al. [7] estimated the causal effect of family size on education, fertility, and income using twin data in Israel, and any evidence of the quantity–quality trade-off was not found. Black et al. [13] used the twin variables as an instrument for family size after controlling for variables including parental education, sibling effects, and birth order, and found that the family size effect did not follow a quantity–quality trade-off relationship. Glick et al. [14] used first-birth twins as a natural experiment to estimate the effect of unplanned parenthood on the children's nutritional status and school enrolment in Romania, and found that first-birth twins had a negative impact on human capital investment in children, especially in the late-birth-order siblings. However, the use of twin status as IV for family size generates considerable controversy. Although the birth of twins can accidently result in an increase in family size, other unobserved factors such as genetics may affect the quality of children, thereby weakening the validity of IV. Meanwhile, the birth of twins may change the future fertility decisions and their attitudes toward human capital investment in children due to short-term economic pressures.

The use of the implementation of family planning policy as a new source of exogenous IV is another approach [15–19]. The policy is unpredictable at the time of introduction and thus has exogenous influences on family size. However, a few countries have pursued family planning policies and different policies have been implemented across regions and over time. In China, a series of family planning policies have been formulated since the 1970s. The one-child policy was introduced in 1979, the two-child policy was introduced in 2015, and the new three-child policy was put forward in 2021. The shift in family planning policy is considered as an exogenous variable on reproductive decision, which is beneficial to overcome the endogeneity of family size. Therefore, the shift in family planning policy provides a natural experimental to test the quantity–quality trade-off model. Census data released by China's National Bureau of Statistics show that a shift in family planning control policies significantly affects the family size. After the implementation of the one-child policy, a decrease in the average family size from 4.41 to 3.10 was observed from 1982 to 2010. The family size increased to 3.35 in 2015 after the implementation of the two-child policy. During the period of family planning policy, China sustained a high rate of annual economic growth, and a high growth in per capita incomes was realized. However, the income gap between families has widened with rapid economic growth in China, as evidenced by the relatively large Gini coefficients in the range of 0.46~0.49 in the past 20 years. The widening income gap between families has also led to disparities in the human capital investment in children, which may affect intergenerational mobility including the intergenerational transmission of occupation [20], poverty [21], economic status [22], etc. Furthermore, the

traditional Chinese ideology of "prioritizing boys over girls" still exists [23–25], and the gender preference of parents may affect their human capital investment decisions. It is estimated that the intergenerational income mobility is relatively low in China [26–28], and high levels of persistence of intergenerational correlation are present. However, it is still not very clear as to what actually determines the degree of intergenerational income mobility by now.

Allowing for variations in family size is one way to learn more about the mechanism underlying intergenerational income mobility, which motivated us to study the intergenerational income mobility with the shift in family planning policies in China. This paper presents an econometric framework to estimate the relative and absolute intergenerational income mobility in a framework that encompasses the family size using the fertility rates allowed by family planning policies as IV for family size. The ordinary least squares (OLS), two-stage least squares (2SLS), and logit regressions were utilized on a robust dataset in this study. OLS regression was conducted to reveal the relationship between family size and intergenerational income mobility. Robustness testing with a 2SLS regression by using IV corroborated the OLS results. Logit regression provides insights into how well family size can affect the probability of upward mobility. Furthermore, the family-size effect on intergenerational income mobility was tested based on the quantity–quality trade-off analysis, which provides a theoretical basis for shaping family planning policies toward sustainability.

The rest of the paper is organized as follows. Section 2 presents empirical frameworks for estimating the absolute and relative intergenerational income mobility by family size. Section 3 describes the data and variables. Section 4 reports the regression results. The family-size effects on intergenerational income mobility are estimated in terms of income rank and number of siblings. The family-size effect is then further tested in accordance with the quantity–quality trade-off model based on the three key factors of budget constraints, credit market imperfections, and parental preference. Section 5 concludes the paper with the theoretical and policy implications.

## 2. Empirical Framework

In the present work, intergenerational income mobility was considered in both relative and absolute terms. To allow for family-size effects, the family size was integrated in the estimation of intergenerational income mobility.

### 2.1. Family Size and Relative Intergenerational Income Mobility

The relative intergenerational income mobility refers to the correlation between the parent's and child's income, which is known as intergenerational income elasticity. The economic literature has made important advances in the measurement of intergenerational income mobility using the Galton–Becker–Solon equation based on the following framework of human capital investment: family $i$ consists of a parent and a child, and the expected utility depends on the parental consumption and human capital investment in the child. The baseline empirical model used for measuring intergenerational income mobility is given in the following regression equation:

$$\ln y_{c,i} = \beta_0 + \beta_1 \ln y_{f,i} + \beta_z Z_i + \varepsilon_i \tag{1}$$

where $y_{c,i}$ denotes the child's income in family $i$; $y_{f,i}$ denotes the parent's income; $Z_i$ is the controlling term; and $\varepsilon_i$ is the disturbance term. $\beta_1$ is the intergenerational income elasticity. By incorporating the family-size variable into the Galton–Becker–Solon equation, the empirical model of relative intergenerational income mobility with respective to family size can be obtained:

$$\ln y_{c,i} = \beta_0 + \beta_1 \ln y_{f,i} + \beta_2 sibs + \beta_3 \ln y_{f,i} * sibs + \beta_z Z_i + \varepsilon_i \tag{2}$$

where *sibs* denotes the family size (i.e., the number of children in family *i*). Intergenerational income elasticity is written as $\beta_1 + \beta_3 * sibs$, where $\beta_3$ denotes the impact of family size on the intergenerational income elasticity. $Z_i$ is the controlling term and $\varepsilon_i$ is the disturbance term. If $\beta_3 < 0$, intergenerational income elasticity decreases with an increase in the family size, indicating an increase in the relative intergenerational income mobility and vice versa. Furthermore, parental education and length of service were introduced as IV in order to test the robustness of the model.

## 2.2. Family Size and Absolute Intergenerational Income Mobility

The absolute intergenerational income mobility refers to the degree to which children move up or down, compared with their parents in absolute terms. This paper examined the impact of family size on the absolute intergenerational income mobility by constructing a binary choice model (logit model), which is represented in formal terms:

$$Mobility_{c,i} = \begin{bmatrix} 1 & \text{Rank}(y_{c,i}) > \text{Rank}\left(y_{f,i}\right) \\ 0 & \text{Rank}(y_{c,i}) \leq \text{Rank}\left(y_{f,i}\right) \end{bmatrix} \tag{3}$$

where $Mobility_{c,i}$ denotes the probability of upward mobility, and $\text{Rank}(y_{c,i})$ and $\text{Rank}(y_{f,i})$ denote the child's income rank and parent's income rank separately. In the present work, both the child's income and parent's income were ranked according to their relative position in the national income distribution, and ranks were divided into five categories on the basis of income levels according to a quantile regression approach. If $\text{Rank}(y_{c,i}) > \text{Rank}(y_{f,i})$, $Mobility_{c,i} = 1$, indicating the upward mobility. If $\text{Rank}(y_{c,i}) \leq \text{Rank}(y_{f,i})$, $Mobility_{c,i} = 0$, indicating the downward mobility. Therefore, the impact of family size on the absolute intergenerational income mobility can be estimated as follows:

$$Mobility_{c,i} = \gamma_0 + \gamma_1 sibs_i + \gamma_z Z_i + v_i \tag{4}$$

where *sibs* denotes the family size; $Z_i$ is the controlling term; and $v_i$ is the random error term. The slope parameter of Equation (4) quantifies the dependency of upward mobility probability on the family size.

## 3. Data and Variables

The data used in this work were obtained from the China Family Panel Survey (CFPS) launched by Peking University. The CFPS project including five waves over the period 2010–2018 collected data from 25 Chinese provinces/municipalities/autonomous regions. The CFPS was designed to collect individual-, family-, and community-level longitudinal data in contemporary China, focusing on the economic as well as the non-economic well-being of the Chinese population, with a wealth of information covering topics such as economic activities, education outcomes, family dynamics and relationships, migration, and health. CFPS adopts multiple-level questionnaires and a panel design to track changes in individuals and households. It uses multi-stage, implicit stratification, and probability proportion to size the sampling methods with a sampling frame that integrates rural and urban populations to obtain a nationally representative sample. The variables are defined as follows:

Income variables. Income of the children and parent are the annual total income including wage income, business income, transfer income, properties income, and other income. The male samples between the ages of 16 and 65 were selected, and the school-age samples were excluded in the present study.

Family-size variables. The respondents' number of siblings were used as the actual family size from the adult questionnaire in CFPS. In order to overcome the endogeneity of family size, fertility rates affected by the implementation of family planning policy were considered as an IV for family size. On one hand, family size is highly correlated with the change in fertility rates induced by the family planning policy. On the other hand,

family planning policies in China have shifted in accordance with the population growth trends, and administrative limits on fertility are different in urban and rural areas. In rural areas, families are allowed to have a second child, if the first-birth is a daughter, and therefore the fertility number was obtained from the following question "how many births does/did the family planning policy in your village allow a couple to have" as reported by the respondents in the CFPS adult questionnaire. However, the one-child restriction remains in force in urban areas. Urban families only have one child, because individuals born after 2015 (the announcement of the universal two-child policy) were not included in the adult samples.

Controlling variables. At the individual level, the main variables include age, years of education, birth order, gender, and household registration. At the family-level, the main variables include the average age of the mother and father, average years of education for the mother and father, and the number of children. Table 1 presents the sample summary statistics of the children and parents. At the region-level, the main variables include the regional economic development level and credit market development. Dummy variables are used for regions having a developed credit market (1 = yes). The credit market development is determined on the basis of the average share of personal loans. In the regions having developed credit market, the average share of personal loans was higher than the national average between 1979 and 2008. The regions having developed a credit market include Beijing, Tianjin, Shanghai, Jilin, Jiangsu, Hubei, Shanxi, Shandong, and Shanxi.

**Table 1.** The sample summary statistics of the main variables.

| Variables | Definition | N | Mean | Std | Min | Max |
|---|---|---|---|---|---|---|
| **Children** | | | | | | |
| yc | Income | 5448 | 27,693.86 | 31,535.08 | 2311 | 840,000 |
| age | Age | 5448 | 24.09 | 4.41 | 16 | 40 |
| edu | Years of education | 5448 | 11.16 | 3.83 | 0 | 22 |
| male | Being male (1 = yes, 0 = no) | 5448 | 0.52 | 0.49 | 0 | 1 |
| urban | Being urban (1 = yes, 0 = no) | 5448 | 0.47 | 0.44 | 0 | 1 |
| bo | Birth order | 5448 | 2.49 | 0.96 | 1 | 8 |
| **Parents** | | | | | | |
| yf | Income | 5448 | 27,006.15 | 27,505.20 | 3500 | 1,040,000 |
| fage | Average age | 5448 | 49.99 | 5.09 | 34 | 60 |
| fedu | Average years of education | 5448 | 7.67 | 4.24 | 0 | 18 |
| sibs | Number of children | 5448 | 1.84 | 1.18 | 1 | 8 |
| ps | Fertility rates allowed by family planning policy | 5448 | 1.72 | 0.82 | 1 | 3 |
| pergdp | Per capita income | 5448 | 49,250.56 | 25,292.97 | 13,119 | 140,211 |
| area | Being developed credit market (1 = yes, 0 = no) | 5448 | 0.46 | 0.50 | 0 | 1 |

## 4. Empirical Analysis

### 4.1. Family-Size Effect on Intergenerational Income Mobility

The baseline empirical model of relative intergenerational income mobility is shown in Equation (1), and the empirical model of relative intergenerational income mobility with respective to family size is shown in Equation (2). In order to estimate the impact of family size on intergenerational income mobility, regression analyses were performed using the OLS and 2SLS methods separately. The regression coefficients are shown in Table 2. Columns 1 and 2 show the estimation results with and without incorporating the family-size effect using the OLS method. The relative intergenerational income elasticity estimated from the baseline empirical model was 0.279, and a lower value (0.2576) was obtained with the incorporation of thee family-size effect. The cross-term coefficient of parental income and family size was significantly negative, showing that the intergenerational income elasticity decreased with an increase in family size. Columns 3 and 4 show the estimation results with and without incorporating the family-size effect using the 2SLS method, and the relative intergenerational income elasticity was 0.415 and 0.4032. In comparison, the

cross-term coefficient of parental income and family size through the 2SLS method was more negative than the OLS method.

**Table 2.** The estimated relative intergenerational income elasticity by family size using the OLS and 2SLS method.

| | OLS | | 2SLS | |
|---|---|---|---|---|
| | **(1)** | **(2)** | **(3)** | **(4)** |
| $lny_f$ | 0.279 *** | 0.306 *** | 0.415 *** | 0.499 *** |
| | (0.0137) | (0.0208) | (0.0342) | (0.0482) |
| sib | — | 0.204 ** | — | 0.192 *** |
| | — | (0.0873) | — | (0.0243) |
| sib * $lny_f$ | — | −0.0484 ** | — | −0.0958 *** |
| | — | (0.0211) | — | (0.00590) |
| age | 1.101 *** | 1.009 *** | 0.0872 *** | 0.0720 *** |
| | (0.135) | (0.134) | (0.0147) | (0.0137) |
| age2 | −0.0226 *** | −0.0205 *** | −0.00200 *** | −0.00164 *** |
| | (0.00268) | (0.00266) | (0.000286) | (0.000266) |
| fage | −0.411 ** | −0.405 ** | 0.0214 | 0.0111 |
| | (0.203) | (0.202) | (0.0227) | (0.0210) |
| fage2 | 0.00443 ** | 0.00436 ** | −0.0000962 | −0.0000133 |
| | (0.00205) | (0.00204) | (0.000229) | (0.000212) |
| bo | −1.042 *** | −1.036 *** | −0.166 *** | −0.153 *** |
| | (0.173) | (0.172) | (0.0210) | (0.0199) |
| male | −1.060 *** | −1.066 *** | −0.0761 *** | −0.0802 *** |
| | (0.110) | (0.109) | (0.0126) | (0.0117) |
| urban | 0.645 *** | 0.465 *** | 0.0710 *** | 0.0596 *** |
| | (0.117) | (0.119) | (0.0138) | (0.0129) |
| lnpergdp | 1.271 *** | 1.076 *** | −0.178 *** | −0.193 *** |
| | (0.181) | (0.182) | (0.0310) | (0.0305) |
| area | −0.373 ** | −0.347 ** | 0.0438 ** | 0.0463 ** |
| | (0.159) | (0.158) | (0.0194) | (0.0184) |
| _cons | −5.449 | −1.889 | −1.155 ** | −0.394 |
| | (4.808) | (4.808) | (0.541) | (0.502) |
| $N$ | 5448 | 5448 | 5448 | 5448 |
| adj. $R^2$ | 0.215 | 0.229 | 0.269 | 0.277 |
| Weak instrument test (F) | — | — | 74.28 | 78.91 |

Note: * $p < 0.1$, ** $p < 0.05$, *** $p < 0.01$. Robust standard errors are reported in parentheses.

### 4.2. Income Heterogeneity and Family-Size Effect on Intergenerational Income Mobility

The above regression represents the correlation between the average parent's income and average child's income, but income heterogeneity exerted a significant impact. The quantile regression method provides a way to estimate the family-size effect on intergenerational income mobility in terms of the child's income rank. The child's income is ranked according to their relative position in the income distribution, and ranks are scaled between 0 and 100. The regression coefficients are shown in Table 3. Although the cross-term coefficient of the parent's income and family size was negative, a significant change in the cross-term coefficient with the income percentile rank of child was present, showing that the impact of family size on the intergenerational income elasticity is sensitive to the child's income rank positions. The cross-term coefficients were almost the same for children whose income at the bottom 10–20th percentile rank and the top 80–90th percentile rank. In comparison, the cross-term coefficient for children whose income at the 30–70th percentile rank was more negative. Therefore, no significant change in the intergenerational income mobility with an increase in family size was observed in the bottom-income and top-income families, while an increase in the intergenerational income mobility with an increase in family size present in middle-income families.

**Table 3.** The estimated relative intergenerational income elasticity by family size using the quantile regression method.

| | (1) | (2) | (3) | (4) | (5) | (6) | (7) | (8) | (9) |
|---|---|---|---|---|---|---|---|---|---|
| | **q10** | **q20** | **q30** | **q40** | **q50** | **q60** | **q70** | **q80** | **q90** |
| $lny_f$ | 0.495 *** | 0.395 *** | 0.360 *** | 0.350 *** | 0.336 *** | 0.271 *** | 0.206 *** | 0.297 *** | 0.288 |
| | (0.111) | (0.044) | (0.041) | (0.027) | (0.031) | (0.034) | (0.033) | (0.029) | (0.028) |
| ps | 0.0691 | 0.0418 | 0.0271 ** | 0.0793 *** | 0.0249 ** | 0.291 ** | 0.321 *** | 0.388 *** | 0.361 *** |
| | (0.131) | (0.147) | (0.012) | (0.031) | (0.011) | (0.119) | (0.118) | (0.105) | (0.129) |
| ps * $lny_f$ | −0.026 | −0.037 | −0.042 *** | −0.046 *** | −0.042 ** | −0.059 *** | −0.022 * | −0.021 | −0.026 |
| | (0.027) | (0.035) | (0.012) | (0.011) | (0.012) | (0.017) | (0.013) | (0.031) | (0.032) |
| controls | Y | Y | Y | Y | Y | Y | Y | Y | Y |
| _cons | 4.208 | 8.041 | −0.360 | −5.860 | −10.53 ** | −12.93 ** | −16.70 ** | −22.90 *** | −25.99 ** |
| | (8.394) | (5.091) | (4.456) | (4.108) | (4.979) | (5.265) | (4.859) | (4.010) | (4.785) |
| N | 5448 | 5448 | 5448 | 5448 | 5448 | 5448 | 5448 | 5448 | 5448 |

Note: * $p < 0.1$, ** $p < 0.05$, *** $p < 0.01$.

### 4.3. Family-Size Effect on Intergenerational Income Mobility under Unequal Family Size

We further tested the family-size effect on the relative intergenerational income mobility when family sizes were unequal. The family size was characterized by the number of siblings. The regression results are shown in Table 4. From the OLS regression results in Columns 1, 3, and 5, the intergenerational income elasticities were 0.300, 0.253, and 0.266 for families with one sibling, two siblings, and more than two siblings, respectively. It was found that the intergenerational income elasticities first fell and then rose with an increase in family size. A similar trend was present from the 2SLS regression results in Columns 2, 4, and 6, but a higher intergenerational income elasticity than OLS regression was observed. Therefore, nonlinearity in the intergenerational transmission of income under unequal family size was observed.

**Table 4.** The estimated relative intergenerational income elasticity under unequal family size.

| | Sibling = 1 | | Siblings = 2 | | Siblings ≥ 3 | |
|---|---|---|---|---|---|---|
| | (1) | (2) | (3) | (4) | (5) | (6) |
| | **OLS** | **2SLS** | **OLS** | **2SLS** | **OLS** | **2SLS** |
| $lny_f$ | 0.300 *** | 0.442 *** | 0.253 *** | 0.350 *** | 0.266 *** | 0.416 *** |
| | (0.0271) | (0.133) | (0.0244) | (0.0443) | (0.0288) | (0.125) |
| age | 0.890*** | 0.0740 ** | 1.280 *** | 0.0648 ** | 0.870 *** | 0.0685 ** |
| | (0.368) | (0.0340) | (0.251) | (0.0290) | (0.248) | (0.0329) |
| age2 | −0.019 *** | −0.017 *** | −0.026 *** | −0.017 *** | −0.017 *** | −0.0014 ** |
| | (0.00733) | (0.00065) | (0.00510) | (0.00056) | (0.00506) | (0.00063) |
| fage | −0.611 | 0.0620 | −0.606 * | −0.0522 | 0.257 | 0.0645 |
| | (0.439) | (0.0617) | (0.360) | (0.0424) | (0.429) | (0.0548) |
| fage2 | 0.00655 | −0.00043 | 0.00653 * | 0.000614 | −0.00228 | −0.00054 |
| | (0.00438) | (0.00059) | (0.00369) | (0.00042) | (0.00433) | (0.00054) |
| bo | −2.063 *** | −0.128 | −1.081 *** | −0.257 *** | −0.346 | −0.0970 ** |
| | (0.342) | (0.0782) | (0.322) | (0.0387) | (0.373) | (0.0483) |
| male | −0.814 *** | −0.099 *** | −1.279 *** | −0.089 *** | −0.879 *** | −0.0841 ** |
| | (0.207) | (0.0326) | (0.188) | (0.0208) | (0.235) | (0.0335) |
| urban | 0.853 *** | 0.0723 ** | 0.198 | 0.0366 | 0.455 ** | 0.0717 ** |
| | (0.265) | (0.0344) | (0.215) | (0.0276) | (0.231) | (0.0320) |
| lnpergdp | 1.092 *** | −0.349 * | 0.814 ** | −0.116 *** | 1.319 *** | −0.249 ** |
| | (0.322) | (0.211) | (0.363) | (0.0426) | (0.363) | (0.0989) |
| area | −0.544 * | 0.0844 | −0.391 | 0.0245 | −0.280 | 0.0840 |
| | (0.308) | (0.0799) | (0.298) | (0.0327) | (0.308) | (0.0518) |
| _cons | 7.004 | −1.729 | 2.397 | 1.352 | −22.13 ** | −2.562 * |
| | (10.41) | (1.461) | (8.815) | (1.055) | (9.945) | (1.374) |

**Table 4.** *Cont.*

| | Sibling = 1 | | Siblings = 2 | | Siblings $\geq 3$ | |
|---|---|---|---|---|---|---|
| | **(1)** | **(2)** | **(3)** | **(4)** | **(5)** | **(6)** |
| | **OLS** | **2SLS** | **OLS** | **2SLS** | **OLS** | **2SLS** |
| *N* | 2615 | 2778 | 1903 | 1422 | 930 | 1248 |
| adj. $R^2$ | 0.251 | 0.249 | 0.209 | 0.205 | 0.178 | 0.176 |
| Weak instrument test (F) | — | 43.16 | — | 47.24 | — | 38.66 |

Note: * $p < 0.1$, ** $p < 0.05$, *** $p < 0.01$.

### 4.4. Family-Size Effect on Intergenerational Income Mobility: Testing the Quantity–Quality Trade-Off

The above analyses show that family size has a significant effect on intergenerational income mobility, which is dependent on the income rank and number of siblings. To understand the family-size effect on intergenerational income mobility, it is essential to test the quantity–quality trade-off of children because the family's economic and social resources become diluted as the variation in family size. According to the classical quantity–quality trade-off model [1,2], this trade-off is determined by three key parameters: budget constraints, credit market imperfections, and parental preference. The budget constraint is correlated with the regional economic development level. The credit market imperfections are more prevalent in the region with less-developed credit market. The three key factors are discussed in the following.

*Budget constraints.* The existence of budget constraints in the family can affect the human capital investment in their children, and the variation in family size further determines the magnitude of human capital investments, which might result in the trade-off between the quantity and quality. In this section, the samples were divided into high- and low-income region according to the per capita income to test the family-size effect on intergenerational income mobility. The regression results are shown in Table 5. From the regression results in Columns 1 through 3 in the high-income region, the cross-term coefficient of the parent's income and family size was not significant through the OLS and 2SLS methods, and there was no correlation between the family size and upward income mobility of the child through logit regression. This might be because the human capital investment in each child did not decrease with an increase in the number of children for the high-income region with loose budget constraints, therefore the quantity–quality trade-off was not obvious. However, the cross-term coefficients of the parent's income and family size through the OLS and 2SLS methods were more negative in Columns 4 and 5 in the low-income region. Meanwhile, logit regression in Column 6 showed that the family-size effect on the upward income mobility of the child is significantly negative with a significance level of 0.01. The reason is that the negative effect of family size on the human capital investment in each child is much stronger for the low-income region with tight budget constraints, thus the quantity–quality trade-off is obvious.

**Table 5.** The estimates for the family-size effect on intergenerational income elasticity across different income regions.

| | High-Income Region | | | Low-Income Region | | |
|---|---|---|---|---|---|---|
| | **(1)** | **(2)** | **(3)** | **(4)** | **(5)** | **(6)** |
| | **OLS** | **2SLS** | **Logit** | **OLS** | **2SLS** | **Logit** |
| $lny_f$ | 0.273 *** | 0.307 *** | — | 0.298 *** | 0.324 *** | — |
| | (0.0443) | (0.0328) | — | (0.0335) | (0.0381) | — |
| Sibs | 0.413 | 0.0415 | −0.0555 | 0.220 ** | 0.343 *** | −0.120 *** |
| | (0.466) | (0.1110) | (0.0366) | (0.102) | (0.0833) | (0.0350) |

**Table 5.** *Cont.*

| | High-Income Region | | | Low-Income Region | | |
|---|---|---|---|---|---|---|
| | **(1)** | **(2)** | **(3)** | **(4)** | **(5)** | **(6)** |
| | **OLS** | **2SLS** | **Logit** | **OLS** | **2SLS** | **Logit** |
| Sibs * $lny_f$ | −0.0866 | −0.0153 | — | −0.0228 * | −0.129 *** | — |
| | (0.171) | (0.0104) | — | (0.0118) | (0.0357) | — |
| age | 1.574 *** | 0.0959 *** | 0.468 *** | 1.096 *** | 0.115 *** | 0.509 *** |
| | (0.184) | (0.0244) | (0.120) | (0.148) | (0.0235) | (0.108) |
| age2 | −0.0296 *** | −0.0022 *** | −0.00798 *** | −0.019 *** | −0.0023 *** | −0.0085 *** |
| | (0.0036) | (0.00049) | (0.00239) | (0.0029) | (0.00046) | (0.00210) |
| fage | −0.200 | 0.00919 | −0.0572 | −0.119 | −0.0174 | −0.525 *** |
| | (0.298) | (0.0407) | (0.195) | (0.232) | (0.0336) | (0.179) |
| fage2 | 0.00249 | 0.000086 | 0.0000976 | 0.00128 | 0.000226 | 0.00516 *** |
| | (0.0029) | (0.00041) | (0.0019) | (0.0023) | (0.00033) | (0.0017) |
| bo | −0.192 | −0.136 *** | 0.0162 | 0.00712 | −0.111 *** | 0.439 *** |
| | (0.188) | (0.0401) | (0.128) | (0.169) | (0.0307) | (0.129) |
| male | −0.797 *** | −0.0621 *** | −0.414 *** | −1.251 *** | −0.0654 *** | −0.501 *** |
| | (0.147) | (0.0239) | (0.0991) | (0.124) | (0.0188) | (0.0982) |
| urban | 0.498 *** | 0.0277 | −0.0162 | 0.513 *** | 0.0596 *** | −0.167 * |
| | (0.162) | (0.0262) | (0.100) | (0.131) | (0.0213) | (0.100) |
| area | −0.0667 ** | −0.0674 *** | 0.0628 *** | 0.786 *** | −0.0667 ** | 0.0399 *** |
| | (0.0286) | (0.0266) | (0.0107) | (0.139) | (0.0286) | (0.0108) |
| _cons | −16.93 ** | −4.148 *** | −9.551 ** | −10.27 * | −4.505 *** | 2.010 |
| | (6.827) | (1.509) | (4.343) | (5.305) | (1.264) | (4.241) |
| *N* | 2179 | 2179 | 2179 | 3269 | 3269 | 3269 |
| adj. $R^2$ | 0.281 | 0.262 | — | 0.205 | 0.225 | — |
| Pseudo $R^2$ | — | — | 0.228 | — | — | 0.235 |
| Weak instrument test (F) | — | 65.72 | — | — | 59.16 | — |

Note: (i) The relative intergenerational income elasticity was estimated by the OLS and 2SLS regression, and the absolute intergenerational income elasticity was estimated by logit regression. (ii) * $p < 0.1$, ** $p < 0.05$, *** $p < 0.01$.

Credit market imperfections. In the presence of credit market imperfections, parents cannot easily borrow money from the market, thus have limited resources for human capital investment in their children (i.e., budget constraints). However, with the development of the credit market, the budget constraints can be relaxed, and an increase in the number of siblings does not lead to a reduction in human capital investment in each child, which may result in weaker evidence of the quantity–quality trade-off. To test the heterogeneity in this trade-off, samples were divided into a region with well-developed credit markets and a region with less-developed credit markets. The regression results are shown in Table 6. In the region with well-developed credit markets, the regression results in Columns 1 through 3 showed that the cross-term coefficient of the parent's income and family size was not significant through the OLS and 2SLS methods, and there was no correlation between the family size and upward income mobility of the child through logit regression. The trend was significantly different in the region with less-developed credit markets, according to the regression results in Columns 4 through 6. The cross-term coefficient of the parent's income and family size through the OLS and 2SLS methods was more negative and the significance level reached 0.01 (Columns 4 and 5). Moreover, the family-size effect on the upward income mobility of children was significantly negative through logit regression in Column 6. The parents were subject to less budget constraints due to the relatively mature credit markets in the region with well-developed credit markets, therefore the quantity–quality trade-off was not obvious. In contrast, aa negative quantity–quality trade-off was pronounced in the region with less-developed credit markets where the credit budget is more constrained.

**Table 6.** The estimates for the family-size effect on intergenerational income elasticity across different income regions regarding credit market development.

| | Region with Well-Developed Credit Markets | | | Region with Less-Developed Credit Markets | | |
|---|---|---|---|---|---|---|
| | **(1)** | **(2)** | **(3)** | **(4)** | **(5)** | **(6)** |
| | **OLS** | **2SLS** | **Logit** | **OLS** | **2SLS** | **Logit** |
| $lny_f$ | 0.375 *** | 0.391 *** | — | 0.411 *** | 0.413 *** | — |
| | (0.0577) | (0.0688) | — | (0.0531) | (0.0565) | — |
| sibs | 0.0785 | 0.0903 | −0.0482 | 0.627 * | 0.268 *** | −0.125 *** |
| | (0.454) | (0.524) | (0.0377) | (0.348) | (0.0750) | (0.0341) |
| Sibs * $lny_f$ | −0.01945 | −0.0312 | — | −0.0456 *** | −0.0885 *** | — |
| | (0.0205) | (0.0919) | — | (0.0154) | (0.0324) | — |
| age | 1.189 *** | 0.120 *** | 0.445 *** | 1.205 *** | 0.120 *** | 0.359 *** |
| | (0.206) | (0.0216) | (0.116) | (0.204) | (0.0206) | (0.108) |
| age2 | −0.0229 *** | −0.00256 *** | −0.00893 *** | −0.0240 *** | −0.0025 *** | −0.0058 *** |
| | (0.00409) | (0.000430) | (0.00229) | (0.00407) | (0.00041) | (0.0022) |
| fage | −0.222 | 0.00412 | −0.495 ** | −0.321 | −0.00328 | −0.600 *** |
| | (0.317) | (0.0322) | (0.199) | (0.281) | (0.0307) | (0.173) |
| fage2 | 0.00194 | 0.0000553 | 0.00494 ** | 0.00333 | 0.000118 | 0.00586 *** |
| | (0.00319) | (0.000325) | (0.00200) | (0.00284) | (0.00031) | (0.00174) |
| bo | −0.953 *** | −0.111 *** | −0.450 *** | −0.564 ** | −0.0979 *** | 0.339 ** |
| | (0.263) | (0.0325) | (0.149) | (0.246) | (0.0301) | (0.141) |
| male | −0.958 *** | −0.102 *** | −0.427 *** | −1.209 *** | −0.100 *** | −0.523 *** |
| | (0.162) | (0.0170) | (0.103) | (0.157) | (0.0163) | (0.0933) |
| urban | 0.983 *** | 0.0431 ** | 0.172 | 0.256 | 0.0403 ** | −0.299 *** |
| | (0.187) | (0.0185) | (0.107) | (0.166) | (0.0177) | (0.0932) |
| lnpergdp | 1.890 *** | −0.136 *** | 1.106 *** | 0.658 ** | −0.114 *** | 0.348 ** |
| | (0.231) | (0.0439) | (0.152) | (0.298) | (0.0396) | (0.148) |
| _cons | −18.50 ** | −0.883 | −3.239 | −3.847 | −0.646 | 6.907 * |
| | (7.390) | (0.862) | (4.711) | (6.749) | (0.786) | (4.106) |
| N | 2506 | 2506 | 2506 | 2942 | 2942 | 2942 |
| adj. $R^2$ | 0.243 | 0.198 | — | 0.174 | 0.172 | — |
| Pseudo $R^2$ | — | — | 0.188 | — | — | 0.179 |
| Weak instrument test (F) | — | 38.79 | — | — | 42.63 | — |

Note: (i) The relative intergenerational income elasticity was estimated by the OLS and 2SLS regression, and the absolute intergenerational income elasticity was estimated by logit regression. (ii) * $p < 0.1$, ** $p < 0.05$, *** $p < 0.01$.

Parental preference. The classical quantity–quality trade-off model builds on the notion that parents are impartial to each child. However, parental preferences for sons over daughters still exist in many families in China due to cultural factors, which may weaken the quantity–quality trade-off. In order to test the hypothesis, samples were divided into the son group and daughter group according to the sex of the siblings, and the two groups of samples were further divided into "same-sex children" and "mixed-sex children" separately. The regression results of sons and daughters are shown in Table 7. In families with only sons (Columns 1 through 3) and families with only daughters (Columns 7 through 9), the cross-term coefficient of the parent's income and family size through the OLS and 2SLS methods as well as the family-size effect on the upward income mobility of the child through logit regression were both significantly negative. Therefore, evidence of a trade-off between the quantity and quality was found in families with same-sex children. However, the trends for the son group and daughter group were different in families with mixed-sex children. It was shown that the cross-term coefficient of the parent income and family size was not significant, and there was no correlation between the family size and upward income mobility through logit regression for the son group, as shown in Columns 4 through 6. As a consequence, this trade-off was not obvious for the son group.

In contrast, the cross-term coefficient of parent income and family size was significant for the daughter group, and the significance level was 0.01 and 0.05 through the OLS and 2SLS regression (Columns 10 and 11). The family-size effect on the upward income mobility of daughters was significantly negative through logit regression (Column 12). As a result, a pronounced trade-off was observed for the daughter group. The existence of the "preference to sons over daughters" phenomenon may be responsible for the sex difference in the intergenerational transmission of income. It is reported that sex preference is prevalent in extended families [29,30]. After the implementation of family planning policy, a decrease in the average family size was present in China. In our research samples, the target sample size was 5448 households, but the maximum sample size of households with more than two siblings was 930. The limitation in our study is characterized by the relatively small sample size of extended families.

**Table 7.** The estimates for the family-size effect on intergenerational income elasticity regarding sibling structures.

| | Sons | | | | | | Daughters | | | | | |
|---|---|---|---|---|---|---|---|---|---|---|---|---|
| | Same-Sex Children | | | Mixed-Sex Children | | | Same-Sex Children | | | Mixed-Sex Children | | |
| | (1) | (2) | (3) | (4) | (5) | (6) | (7) | (8) | (9) | (10) | (11) | (12) |
| | OLS | 2SLS | Logit | OLS | 2SLS | Logit | OLS | 2SLS | Logit | OLS | 2SLS | Logit |
| $\ln y_f$ | 0.334 *** | 0.356 *** | — | 0.202 *** | 0.232 *** | — | 0.195 *** | 0.216 *** | — | 0.125 *** | 0.198 *** | — |
| | (0.061) | (0.062) | — | (0.004) | (0.0035) | — | (0.073) | (0.015) | — | (0.011) | (0.018) | — |
| sibs | 0.604 *** | 0.625 *** | −0.159 ** | 0.141 | 0.153 | −0.0615 | 0.607 *** | 0.120 *** | −0.458 *** | 1.395 *** | 1.281 *** | −0.168 * |
| | (0.205) | (0.204) | (0.081) | (0.182) | (0.181) | (0.118) | (0.204) | (0.0337) | (0.112) | (0.338) | (0.465) | (0.012) |
| sibs * $\ln y_f$ | −0.049 ** | −0.049 ** | — | −0.027 | −0.0237 | — | −0.049 ** | −0.019 *** | — | −0.131 *** | −0.115 ** | — |
| | (0.0243) | (0.0242) | — | (0.0197) | (0.0186) | — | (0.0241) | (0.0044) | — | (0.037) | (0.057) | — |
| controls | Y | Y | Y | Y | Y | Y | Y | Y | Y | Y | Y | Y |
| cons | −4.360 | −0.646 | 2.658 | −5.052 | −5.012 | −2.752 | −1.736 | 1.546 *** | 11.72 ** | 23.42 | 22.62 | −6.553 |
| | (7.570) | (0.786) | (9.845) | (7.904) | (7.814) | (13.99) | (7.525) | (0.582) | (5.317) | (14.38) | (14.79) | (9.519) |
| $N$ | 765 | 708 | 765 | 707 | 628 | 707 | 679 | 654 | 679 | 682 | 680 | 682 |
| adj. $R^2$ | 0.188 | 0.183 | — | 0.231 | 0.219 | — | 0.231 | 0.224 | — | 0.267 | 0.266 | — |
| Pseudo $R^2$ | — | — | 0.179 | — | — | 0.211 | — | — | 0.187 | — | — | 0.211 |
| Weak instrument test (F) | — | 58.79 | — | — | 47.34 | — | — | 67.28 | — | — | 54.39 | — |

Note: (i) The relative intergenerational income elasticity was estimated by OLS and 2SLS regression, and the absolute intergenerational income elasticity was estimated by logit regression. (ii) * $p < 0.1$, ** $p < 0.05$, *** $p < 0.01$.

Based on the above testing on the relationship between quantity and quality, evidence of a quantity–quality trade-off can disappear if one of the presumptions does not hold. The family-size effect on the quantity–quality relationship demonstrates great heterogeneity across external factors (economic growth and credit market development) and internal factors (family structures and gender equality). An in-depth analysis of the theoretical mechanism needs further investigation. The present work provides a theoretical basis for shaping the family planning policies toward sustainability to improve the intergenerational income mobility.

## 5. Conclusions

Economists have been trying to reveal the relationship between the quantity and quality of children since the emergence of the household's human capital investment theory. This argument has become one of the important factors for family planning policies in many countries. However, empirical studies aiming to test the quantity–quality trade-off are far from sufficient. This paper examined the effect of family size on intergenerational income mobility by the incorporating of family-size effect into the baseline Galton–Becker–Solon equation. On this basis, the causal relationship between the quantity and quality of children was tested to explain the family-size effect on intergenerational income mobility.

In this paper, intergenerational income elasticity tended to decrease with an increase in family size. The impact of family size on intergenerational income elasticity was sensitive to the income rank positions, exhibiting no significant change with family size in the bottom- and top-income families, but an increase in the intergenerational income mobility with family size in middle-income families. Estimation under an unequal family size showed that intergenerational income elasticities first fell and then rose with an increase in family size. The quantity–quality trade-off analysis was applied to further test the family-size effect on intergenerational income mobility. Obvious family-size effects on intergenerational income mobility were observed in the low-income region with tight budget constraints and in the region with less-developed credit markets, this trade-off was obvious. Pronounced family-size effects on intergenerational income mobility were found in families with same-sex children. In families with mixed-sex children, family size did not produce a significant effect on the intergenerational income mobility for the son group, while family size had a significant effect for the daughter group. The existence of a "preference for sons over daughters" phenomenon may be responsible for the sex difference in the intergenerational transmission of income. In other words, China's family planning policies induce a decrease in the average family size, and diminish the associations in income between parents and children, which can promote equal opportunities in society. However, a strong association in income between parents and sons was observed in families with mixed-sex children due to cultural factors in China.

Our findings not only provide useful evidence with respect to the quantity–quality trade-off theory by adopting an intergenerational approach, but are also helpful in understanding the human capital investment in family. Family planning policies have been widely adopted in developing countries such as China, India, Brazil, Mexico, Kenya, and Bangladesh. This work provides implications for future family planning policies in the sustainable development goal in developing countries with large populations.

**Author Contributions:** Conceptualization, X.M.; Methodology, X.M. and S.C.; Software, X.M.; Validation, X.M. and S.C.; Formal analysis, X.M.; Investigation, X.M.; Resources, S.C.; Data curation, X.M.; Writing—original draft preparation, X.M.; Writing—review and editing, S.C.; Supervision, X.M.; Funding acquisition, X.M. All authors have read and agreed to the published version of the manuscript.

**Funding:** This study was supported by the National Social Science Foundation of China (grant no. 20CSH034).

**Institutional Review Board Statement:** Not applicable.

**Informed Consent Statement:** Not applicable.

**Data Availability Statement:** Data accessed on 1 September 2020 from China Family Panel Survey at http://www.isss.pku.edu.cn/cfps/en/index.htm.

**Conflicts of Interest:** The authors declare no conflict of interest.

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
