# Peer review of "Family-Size Effect on Intergenerational Income Mobility under China’s Family Planning Policy: Testing the Quantity–Quality Trade-Off"

_sustainability, doi:10.3390/su141912559_

Round 1
Reviewer 1 Report
· This study aims to estimate the family-size effect on intergenerational income mobility and examine the trade-off between the quantity and quality of children. This research can contribute to understanding family investment in human capital as a basis for designing sustainable family planning policies. This study has strengths, especially in using controlling variables, not only at the individual level but also at the family and regional levels.
The trade-off between quantity- quality and the importance of sustainable family planning programs is more prominent in developing countries than in developed countries. However, references related to studies in developing countries are still relatively limited. The author is expected to explore further related studies in developing countries, especially in countries with large populations such as Indonesia.
The operational definition of variables related to the parent variable still needs to be clarified. Age and years of education from parents. Is it just the father, the mother, or the average of both (father and mother)?
In the data and variables subsection, the two main variables at the regional level are economic growth and credit market development. But the model used is per capita income, not economic growth. Credit market development is used as a dummy variable, but there is no explanation based on measurement and categorization between criteria 1 (yes) and criteria 0 (no).
There is no logical explanation for using credit market development as an economic variable at the regional level. Instead of using the wage variable (which is directly related to income), the author prefers to use the Credit market development variable.
The author needs to describe the basis for using OLS, 2SLS, and logit regression simultaneously in estimating the model. In addition, the author should be able to show that these three models can complement and sharpen the analysis. However, the author sees more differences in each method's direction and coefficient values ​​in the study, so the reader will find it difficult to get a complete conclusion from his findings.
The conclusions given are appropriate and consistent with the evidence and arguments presented. However, the authors have not disclosed both theoretical and theoretical recommendations.
Author Response
Dear Reviewer 1:
Thank you very much for your valuable comments concerning our revised manuscript, which are very helpful for our work. We have carefully considered the comments and revised the manuscript in details as below, and mark the changes by red in the revised manuscript.Reviewers' comments:
This study aims to estimate the family-size effect on intergenerational income mobility and examine the trade-off between the quantity and quality of children. This research can contribute to understanding family investment in human capital as a basis for designing sustainable family planning policies. This study has strengths, especially in using controlling variables, not only at the individual level but also at the family and regional levels.
- The trade-off between quantity-quality and the importance of sustainable family planning programs is more prominent in developing countries than in developed countries. However, references related to studies in developing countries are still relatively limited. The author is expected to explore further related studies in developing countries, especially in countries with large populations such as Indonesia.
Authors’ response: Thanks for your suggestion. The related studies is developing countries including Indonesia, Senegal and Uganda are provided, as shown in page 2, line 54: “In Senegal and Uganda, the existence of a quantity-quality trade-off was found for most of the quality attributes covering education, nutrition and health care [10]. In Indonesia, the statistically meaningful evidence of a quantity-quality trade-off at select quantiles in the tails of the distribution for body mass index was found [11].”
[10] Hoyweghen, K.V.; Bemelmans J.; Feyaerts H., et al. Small family, happy family? fertility preferences and the quantity-quality trade-off in Sub-Saharan Africa, 2022. (Preprint)
[11] Millimet, D. L.; Wang, L. Is the quantity-quality trade-off a trade-off for all, none, or some?. Econ. Dev. Cult. Change, 2011, 60, 155-195.
- The operational definition of variables related to the parent variable still needs to be clarified. Age and years of education from parents. Is it just the father, the mother, or the average of both (father and mother)?
Authors’ response: The controlling variables including age and years of education of parents are clealy defined in the text and Table 1, as shown in page 5, line 208: “At the family-level, the main variables include average age of mother and father, average years of education for mother and father, and number of children.”.
- In the data and variables subsection, the two main variables at the regional level are economic growth and credit market development. But the model used is per capita income, not economic growth. Credit market development is used as a dummy variable, but there is no explanation based on measurement and categorization between criteria 1 (yes) and criteria 0 (no).
Authors’ response: (i) We are sorry for the mistake. The two main variables at the regional level are regional economic development level and credit market development. Therefore, per capita income is used in the model. The sentence is revised, as shown in page 5, line 211: “At the region-level, the main variables include regional economic development level and credit market development.”
(ii) Credit market development is used as a dummy variable, which is further explained as shown in page 5, line 212: “Dummy variables are used for regions having developed credit market (1 = yes). The degree of credit market development is determined on the basis of average share of personal loans, and the average share of personal loans is higher than the national average in the regions having developed credit market between 1979-2008. The regions having developed credit market include Beijing, Tianjin, Shanghai, Jilin, Jiangsu, Hubei, Shanxi, Shandong and Shanxi.”
- There is no logical explanation for using credit market development as an economic variable at the regional level. Instead of using the wage variable (which is directly related to income), the author prefers to use the Credit market development variable.
Authors’ response: Yes. The wage variable is a better economic variable at the regional level. In the present study, we pay attention to the economic variable, which may be an important factor in diluting the quantity-quality trade-off. In Section 4.4, the preference of credit market development variable is clarified, as shown in page 8, line 278: “According to the classical quantity-quality trade-off model [1,2], the quantity-quality trade-off is determined by three key parameters: budget constraints, credit market imperfections and parental preference. The budget constraint is correlated with the regional economic development level. The credit market imperfections are more prevalent in the region with less-developed credit market. The three key factors are discussed in the following.”.
- The author needs to describe the basis for using OLS, 2SLS, and logit regression simultaneously in estimating the model. In addition, the author should be able to show that these three models can complement and sharpen the analysis. However, the author sees more differences in each method's direction and coefficient values in the study, so the reader will find it difficult to get a complete conclusion from his findings.
Authors’ response: Thanks for your suggestions. The three estimation approaches (OLS, 2SLS and logit regression) is further explained in the Introduction, as shown in page 3, line 112: “The ordinary least squares (OLS), two-stage least squares (2SLS), and logit regressions are utilized on a robust dataset in this study. OLS regression is conducted to reveal the relationship between family size and intergenerational income mobility. Robustness testing with a 2SLS regression by using instrumental variables corroborates the OLS results. Logit regression provides insights into how well the family size can affect the upward mobility probability.”.
- The conclusions given are appropriate and consistent with the evidence and arguments presented. However, the authors have not disclosed both theoretical and theoretical recommendations.
Authors’ response: Yes. The theoretical recommendations and policy implication are provided in Section 5, as shown in page 13, line 404: “Our findings not only provide useful evidence with respect to the quantity–quality trade-off theory by adopting an intergenerational approach, but also are helpful to understand the family investment in human capital. Family planning policies have been widely adopted in developing countries such as China, India, Brazil Mexico, Kenya and Bangladesh. This work provides an implication for future family planning policies to the sustainable development goal in those developing countries with large populations.”.
Reviewer 2 Report
Thank you for providing me with this opportunity to see your work. Your data and your findings are interesting and seem reasonable. They concern an issue of interest to people everywhere, including China. Recent opinion surveys (PEW Research) suggests that parents in many countries are pessimistic about the furure financial welfare of their children. You must let some native English speaker or other competent person knowledgeable about economics improve the language. Often you put in "the" where it doesn't belong, and the opposite etc. It would also be good to have a few lines at least in the abstract and the conclusion in simple English about the results, for readers who are not economists, and who don't know what elasticity etc. is... That may give you more readers. I saw two articles in The Economist (19 Sept 2020 The landlords are back resp. 11 June 2022 Class revival), about the long arm of the family, that is, how children and/or grand-children of the old elite manages to climb socially in China. No, they didn't say anything about family size... Thank you again, and Good Luck!
Author Response
Dear Reviewer 2:
Thank you very much for your valuable comments concerning our revised manuscript, which are very helpful for our work. We have carefully considered the comments and revised the manuscript in details as below, and mark the changes by red in the revised manuscript.Reviewers' comments:
- Thank you for providing me with this opportunity to see your work. Your data and your findings are interesting and seem reasonable. They concern an issue of interest to people everywhere, including China. Recent opinion surveys (PEW Research) suggests that parents in many countries are pessimistic about the future financial welfare of their children. You must let some native English speaker or other competent person knowledgeable about economics improve the language. Often you put in "the" where it doesn't belong, and the opposite etc. It would also be good to have a few lines at least in the abstract and the conclusion in simple English about the results, for readers who are not economists, and who don't know what elasticity etc. is... That may give you more readers. I saw two articles in The Economist (19 Sept 2020 The landlords are back resp. 11 June 2022 Class revival), about the long arm of the family, that is, how children and/or grand-children of the old elite manages to climb socially in China. No, they didn't say anything about family size... Thank you again, and Good Luck!
Authors’ response: Thanks for your suggestion. The manuscript is polished to make it more readable. I have read the recommended articles in The Economist. I do not know much about the situation as described in the articles. In my opinion, the real situation in China is not consistent with the article. Anyway, I would like to tell you the real situation in China through Email ([email protected] or [email protected]) if you like. Meanwhile, I would like to invite you to come to China. I also Good Luck!
- Line 43, please correct Swiss into Sweden.
Authors’ response: We are sorry for the mistake. We have corrected Swiss into Sweden. Thank you!

Reviewer 3 Report
1. Research design of the CFPs (2010-2018) should be well explained. What is the research design for each survey; including sampling design, assurance of representativeness etc. Did they all have similar research design? Do they have common parameters that may them comparable or to what extent are they comparable?
2. The effects of sex preference on quality-quantity tradeoff is often similar to that of extended family system both of which are culture based. Your frameworks and models were largely based on a nucleated system. The extent to which extended family system is common in China will determine how weighty extended family system is as a limitation to your framework. Your work will be strengthened by discussing such limitations.
3. The manuscript need a conceptual framework, and empirical framework, to give more clarity to the manuscript and enable readers understand the sequence of relationships tested, and the results presented.
4. empirical analysis should distinguish models for absolute vs. those for relative inter-generational income mobility. If possible, the tables should be labelled as such to distinguish the two perspectives.
5. The entire manuscript need editorial work especially the introduction section.

Author Response
Dear Reviewer 2:
Thank you very much for your valuable comments concerning our revised manuscript, which are very helpful for our work. We have carefully considered the comments and revised the manuscript in details as below, and mark the changes by red in the revised manuscript.Reviewers' comments:
- Research design of the CFPs (2010-2018) should be well explained. What is the research design for each survey; including sampling design, assurance of representativeness etc. Did they all have similar research design? Do they have common parameters that may them comparable or to what extent are they comparable?
Authors’ response: Thanks for your suggestion. Research design, implementation and data quality are provided, as shown in page 4, line 180: “The CFPS is designed to collect individual-, family-, and community-level longitudinal data in contemporary China, focusing on the economic, as well as the non-economic, wellbeing of the Chinese population, with a wealth of information covering such topics as economic activities, education outcomes, family dynamics and relationships, migration, and health. The CFPS adopts multiple-level questionnaires and a panel design to track changes in individuals and households. It uses multi-stage, implicit stratification, and probability proportion to size sampling methods with a sampling frame that integrates rural and urban populations to obtain a nationally representative sample.”
- The effects of sex preference on quality-quantity tradeoff is often similar to that of extended family system both of which are culture based. Your frameworks and models were largely based on a nucleated system. The extent to which extended family system is common in China will determine how weighty extended family system is as a limitation to your framework. Your work will be strengthened by discussing such limitations.
Authors’ response: Thanks for your nice suggestion. The sex difference phenomenon in the intergenerational transmission of income is discussed considering the family system, as shown in page 11, line 357: “It is reported that sex preference is prevalent in extended families [29, 30]. After the implementation of the family planning policy, a decrease in the average family size is present in China. In our research samples, the target sample size is 5448 households, but a maximum sample size of households with more than three siblings is 930. The limitation in our study is characterized by the relatively small sample size of extended families.”.
- The manuscript need a conceptual framework, and empirical framework, to give more clarity to the manuscript and enable readers understand the sequence of relationships tested, and the results presented.
Authors’ response: The conceptual framework is provided at the end of Introduction, as shown in as shown in page 3, line 121: “The paper proceeds as follows: Section 2 presents the empirical frameworks for estimating the absolute and relative intergenerational income mobility by family size. Section 3 describes the data and variables. Section 4 reports the regression results. The family size effects on the intergenerational income mobility are estimated in terms of income rank and number of siblings. The family-size effect is further tested in accordance to the quantity-quality trade-off model based on the three key factors of budget constraints, credit market imperfections and parental preference. Section 5 concludes the paper with theoretical and policy implication.”
- empirical analysis should distinguish models for absolute vs. those for relative intergenerational income mobility. If possible, the tables should be labelled as such to distinguish the two perspectives.
Authors’ response: The absolute or relative intergenerational income mobility are shown in the table captions or the notes.
- The entire manuscript need editorial work especially the introduction section.
Authors’ response: The manuscript is polished to make it more readable.

Round 2
Reviewer 2 Report
Dear authors,
Thank you for the revised manuscript, where you have added some new references. I have mostly linguistic suggestions to improve the text. English is an easy language, but indeed difficult to be perfect in. I note an overuse of "the" where it is better without, and some other issues.
265 Better: "more than two siblings" (or "three or more siblings"). It would also be interesting to know how many in your population that have No sibling, one sibling, two siblings, and three or more siblings. (Or maybe you say, and I missed it?)
279 too many quality-quantity trade-offs... I would suggest you just mention "this trade-off" the second time. Will be understood.
375 "the theoretical basis" sounds too ambitious, maybe "a theoretical basis", or (better) "a contribution to a therotical basis", humbler.
408 need more commas: "Brazil, Mexico, Kenya, and Bxx". Commas in enumerations.
I would also suggest that you write a short sentence (or two) in simple English for the abstract (if space enough) and for the conclusion about the results, for potential readers not familiar with advanced statistics.
Good Luck with your work!
Author Response
Dear Reviewer 2:
Thanks for your nice suggestion. Please see the attched file. Good Luck to you.
Best regards,
Shenghu Chen, Xinxin Mu
